# The Rising of Paleontology in China: A Century-Long Road

**DOI:** 10.3390/biology11081104

**Published:** 2022-07-25

**Authors:** Zhonghe Zhou

**Affiliations:** Institute of Vertebrate Paleontology and Paleoanthropology, Chinese Academy of Sciences, 142 Xizhimenwai Dajie, Beijing 100044, China; zhouzhonghe@ivpp.ac.cn

**Keywords:** paleontology, China, history, 20th century, 21th century

## Abstract

**Simple Summary:**

A brief account of the history of Chinese paleontology for about one century is provided with some perspectives for its future development. The development of Chinese paleontology is closely related to its social-ecological background as well as its connection to the outside world. On the other hand, the rising of the Chinese paleontology also benefitted from its rich fossil resources as well as the integration with other biological and geological disciplines and the use of new technologies.

**Abstract:**

In this paper, the history of paleontology in China from 1920 to 2020 is divided into three major stages, i.e., 1920–1949, 1949–1978, and 1979–2020. As one of the first scientific disciplines to have earned international fame in China, the development of Chinese paleontology benefitted from international collaborations and China’s rich resources. Since 1978, China’s socio-economic development and its open-door policy to the outside world have also played a key role in the growth of Chinese paleontology. In the 21st century, thanks to constant funding from the government and the rise of the younger generation of paleontologists, Chinese paleontology is expected to make even more contributions to the integration of paleontology with both biological and geological research projects by taking advantage of new technologies and China’s rich paleontological resources.

## 1. Introduction

This paper aims to provide a brief account of the history of the development of paleontology in China. The year 1920 is chosen as an important starting point for Chinese paleontology. During the past century from 1920 to 2020, Chinese paleontology has gone through several difficult periods and was only able to usher in the rapid growth seen over the past several decades thanks to the continuous efforts of the first-generation paleontologists in the 20th century to the youngest generation in the 21th century. In a chronological approach, this paper also intends to reflect the major characteristics of the discipline at various stages of the history with a brief introduction of the major achievements and an overview of how Chinese paleontologists have benefitted from international collaborations and the social-economic changes in China, focusing in particular on those since 1978. Finally, this paper discusses a few brief predictions regarding the future directions and challenges for Chinese paleontology.

## 2. Before 1920—The Collecting and Studying of Chinese Fossils by Foreigners

In China, fossils have been recognized as the remains of ancient animals, plants and fungi as well as seen as evidence of changes of the Earth’s environment for over two thousand years. However, they had never been studied by Chinese scholars as a scientific subject until modern sciences (e.g., geology, paleontology) began to be introduced to China during the 19th century. The first scientific collection of Chinese fossils was made by western explorers, missionaries, geographers, and geologists, particularly gaining momentum since the opium war in 1940, and then studied by western paleontologists from Europe and the Americas. For instance, L. de Korunck published on two brachiopods from China in 1943, and R. Owen published the first paper on Chinese mammal fossils [1]. E. Koken and M. Schlosser published books on Chinese fossil mammals in 1885 and 1903, respectively, which represent the earliest systematic studies on Chinese vertebrate fossils [2,3].

Beginning in the 1860s, French missionary P.A. David started to collect fossil fish from the Mesozoic lake deposits in western Liaoning. These fossils were later named as *Lycoptera* and are regarded as one of the typical elements of the Jehol Biota, which is now best known for producing many exceptional feathered dinosaurs, birds, and mammals. American geologist and explorer R. Pumpelly was one of the first western scientists to carry out geological surveys in China. He collected many fossils from China from 1863~1865 and proposed the lacustrine facies of the Chinese loess [4]. German scientist F. Richthofen made extensive geographic and geological explorations in China during 1868 and 1872. He also collected many invertebrate fossils as well as recorded their stratigraphic data in the majority of Chinese provinces.

During the late 19th century and early 20th century, western explorers and scholars also made many scientific expeditions to China. Swedish geographer and explorer S.A. Hedin made several expeditions to western China starting in 1890, and the ancient city of Loulan (Kroraina) in Xinjiang was one of his major discoveries during his voyage across the Taklamakan Desert. Also notable is the Central Asian Expeditions organized by the American Museum of Natural History in 1916 and 1919, which resulted in the discovery of many fossil mammals in Inner Mongolia. In the early 20th century, Russian geologists collected many dinosaur fossils from Heilongjiang, northeastern China, including some duck-billed dinosaurs.

Before 1920, there were hardly any Chinese scientists known mainly as a paleontologist. The first Chinese scientist who published his research on paleontology was Rongguang Qi, who was among the first group of teenagers sent to America by the Qing Imperial Dynasty government in 1871. He was trained in geology and mineral resources. In 1910, he published his study on invertebrates and plants from Hebei, North China.

The first attempt to teach college-level geology in China started in 1909 at the Imperial University of Peking (known as Peking University since 1912). In 1914, Wenjiang Ding (V.K. Ting; 1887–1936), a 1911 graduate of the University of Glasgow, started to teach China’s first college-level course in paleontology at the Geological Institute established in 1913. This produced the first generation of domestically trained geologists, including a few paleontologists who were then hired by the National Geological Survey of China that was officially founded in 1916 with Ding as its first director. The National Geological Survey of China has also been generally regarded as the first national scientific institution with an international reputation in Chinese history. Among many of his tremendous achievements, Ding also made contributions to many pioneering works in Chinese paleontology and stratigraphy. He discovered the first Devonian fish in Yunnan in 1913 and pioneered the study of fossil plants in China.

In summary, upon entering the 20th century, China had gone through major social changes, and its door began to open to the outside world. In 1911, the Imperial Qing Dynasty was ended and replaced by the Republic of China. A few more young Chinese were sent to Western countries to learn science and technology. The early 20th century was the preparation stage for the early development of paleontology in China [5].

## 3. From 1920 to 1949—Founding Stages of Chinese Paleontology

1920 was a turning point for Chinese paleontology. By the invitation of Wenjiang Ding, two prominent western-trained geologists and paleontologists joined Peking University. One of them was A.W. Grabau from Columbia University. Grabau not only taught at Peking University, but was also the Chief Paleontologist at the Chinese Geological Survey (Figure 1). The second was Siguang Li who graduated from Birmingham University with a master’s degree in 1919 (Ph.D. degree in 1931). During 1920–1937, Grabau, Li, and their colleagues mentored and trained many of the first generation of geologists and paleontologists in China who would later become the leading figures in the majority of disciplines in Chinese geology and paleontology. It is worthy to note that Grabau has been also well known for proposing the Jehol Fauna and Jehol Series in western Liaoning, northeastern China [6,7]. He also named the Changxing limestone in Zhejiang Province, where two GSSPs (Global Boundary Stratotype Sections and Points) were later established, including the boundary between the Permian and Triassic.

Many of the first-generation Chinese paleontologists had been trained overseas and earned their Ph.D. degrees during the 1920s–1940s before they returned to China to pioneer their research fields. Among them were the invertebrate paleontologists Yunzhu Sun and Senxun Yue, vertebrate paleontologist Zhongjian Yang (Chung-Chien Young), and paleobotanist Xingjian Si, who received their Ph.D. degrees in Germany in 1927, 1936, 1928, and 1933, respectively. Zanxun Yin, geologist and paleontologist, and Wenzhong Pei, paleoanthropologist and archaeologist, earned their Ph.D. degrees from France in 1931 and 1937, respectively. Invertebrate paleontologists Jianzhang Yu and Hongzhen Wang received their Ph.D. degrees from England in 1935 and 1947. Zunyi Yang, an invertebrate paleontologist, received his Ph.D. degree from the USA in 1939. They have all became the backbones of Chinese paleontology after they returned to China.

It is notable that scientific expeditions organized by organizations of Western countries continued to be active in China during the 1920s and 1930s. For instance, the Central Asiatic Expeditions led by R.C. Andrews of the American Museum of Natural History during 1922 and 1930 resulted in the discovery of many dinosaur skeletons and dinosaur eggs in Inner Mongolia as well as many Mesozoic and Cenozoic mammals, including some of the earliest placental mammals. However, with the appearance of the first generation of Chinese paleontologists, international collaborations for the study of Chinese paleontology and stratigraphy became more and more prominent and productive. For instance, the Sino-Swedish Expedition which carried out multi-discipline scientific research in north and northwest China from 1927–1935 was co-led by S. Anders and Fuli Yuan, a geologist from Peking University, and many fossil reptiles and mammal fossils were collected during these field excursions. Zhongjian Yang and Wenzhong Pei joined the Central Asiatic Expedition in 1930.

One of the most successful collaborations between the first generation of Chinese paleontologists and their Western colleagues is probably the excavation of the fossil humans in Choukoudian (Figure 2) as well as the study on Cenozoic vertebrates and their stratigraphy. The Swedish geologist J.G. Andersson was invited by the Chinese government to act as consultant for the mining industry from 1914 to 1924. He discovered the Choukoudian site in 1918. During 1921–1922, Andersson assigned the Austrian paleontologist O. Zdansky to excavate at Choukoudian, which resulted in the discovery of two human teeth in addition to an abundance of mammal fossils. In 1927, with the support from the Rockefeller Foundation, D. Black from the Beijing Union Medical College Hospital (BUMCH) was able to continue the excavation of Choukoudian and study the Cenozoic cave deposits based on an agreement with the Geological Survey of China. In 1929, Wenzhong Pei led the excavation in Choukoudian and discovered the first skull of the Peking Man (*Homo erectus*), which drew some international attention for Chinese paleontology and paleoanthropology.

The collaboration between Black and the National Geological Survey of China also resulted in the founding of the Cenozoic Research Laboratory of the Geological Survey of China in 1929, which was the predecessor of the Institute of Vertebrate Paleontology and Paleoanthropology (IVPP). Black was appointed as the honorary head of the lab with Zhongjian Yang as the deputy head and P. T. de Chardin as the adviser. After Black died in 1925, his position was filled by German paleontologist F. Weidenreich. Clearly, the Cenozoic laboratory was designed and intended to be an international research unit.

In 1922, with the support of Grabau and Anderson, Ding founded the first Chinese paleontological journal *Palaeontologia Sinica*, and he became the chief editor of this journal for nearly 15 years. The journal published only English and German papers in the beginning, but later published Chinese papers as well. The majority of the important Chinese paleontological and stratigraphic studies during the 1920s and 1930 were published in this journal.

In 1928, the National Geological Survey of China formally established its Paleontology Laboratory. In the same year, the National Research Institute of Geology was founded in Nanjing with its own Stratigraphy and Paleontology Laboratory, representing the growth of paleontological research being conducted in China.

The paleontological and stratigraphic studies carried out by Chinese paleontologists started to grow in number quickly during the 1920s and 1930s. To list a few examples, Xichou Tan excavated dinosaurs in Laiyang, Shandong Province in 1923 and collected many dinosaur bones, some of which were later published and named as a duck-billed dinosaur *Tanius sinensis* [8]. In addition, a large number of fish, insect, and plant fossils were also collected. In 1923, Zanheng Zhou, a graduate from the National Research Institute of Geology in 1916 who also later trained in Sweden, published his study on the fossil plants from Shandong Province [9], representing the first paleobotanic paper by a Chinese paleontologist.

The book “Contributions to Cambrian Faunas of North China” written by Yunzhu Sun [10] marked the first paleontological book by a Chinese paleontologist. Some domestically educated paleontologists also did great work in paleontology and biostratigraphy. For instance, Yazhen Zhao, a graduate from Peking University was hired by the National Geological Survey of China in 1923. During his short but glorious career, he published several classic books on the study of brachiopods and stratigraphy before he was tragically killed by a bandit while in the field in 1929 at the age of 31.

Siguang Li published his classic monography “Fusulinidae of North China” in 1927 [11]. In 1927, Zhongjian Yang published the book “Fossil Rodents from North China” [12] based on his doctoral dissertation, which was the first monography on Chinese vertebrate paleontology. Zhongjian Yang became the father of Chinese vertebrate paleontology. His earlier work focused on Zhoukoudian and other Late Cenozoic mammalian faunas of northern China. After 1938, his main research interest shifted to Mesozoic dinosaurs and synapsids. His collaborations with P. Teilhard de Chardin were very successful and contributed greatly to the establishment of the geochronology and the stratigraphic succession of the Tertiary and Quaternary periods in China.

Xingjian Si (Hsing-Chien Sze) came back to China after he was awarded a Ph.D. degree for his research on Chinese fossil plants [13] and contributed greatly to the study of Paleozoic and Mesozoic plants in China, which earned him international fame.

Thanks to the efforts of Zhongjian Yang and Yunzhu Sun, the Paleontological Society of China was founded in 1929. Yunzhu Sun elected the first president, and he was elected the vice chairman of the International Paleontological Association in 1948.

Although the 1920s and 1930s witnessed the rapid growth of paleontology in China, starting in 1937 China entered the Second Sino-Japanese War that ended in 1945 and then the Chinese civil war that ended in 1949, which resulted in the founding of the People’s Republic of China. Paleontological study during these years became difficult. The National Geological Survey of China moved to Yunnan. Peking University also moved to Yunnan to merge with Tsinghua University and Nankai University, and together they became the National Southwestern Associated University. At this new conglomerate University, students continued to be taught geology and paleontology, and many of the graduates later became the some of the most important figures in Chinese paleontology (e.g., Hongzhen Wang, Enzhi Mu, Dongsheng Liu, Zhiwei Gu, Yichun Hao etc.). As a 1937 graduate from Peking University, Yanhao Lu, a Cambrian and trilobite expert, first worked in the National Southwestern Associated University and then moved to the National Geological Survey of China. In reality, Chinese paleontological study had not completely stopped. For instance, Zhongjian Yang continued his work on the Lufeng dinosaur fauna in Yunnan during the trying years from 1938 to 1945 and had discovered the Jurassic dinosaur *Lufengosaurus* Fauna [14]. *Lufengosaurus* also represents China’s first dinosaur with a mounted and complete skeleton. In 1940, Young was appointed the head of the Laboratory of Vertebrate Palaeontology and the honorary head of the Cenozoic Department of the National Geological Survey of China. In addition to the work he did in Yunnan, he also carried out fieldwork in Northwestern China. During the same time, Zanxun Yin and Yanhao Lu studied the Paleozoic biostratigraphy in southwestern China.

In sum, after the fall of Qing Dynasty in the 1911 and before the Japanese invasion in 1937, China had witnessed a relatively fast period of economic growth and the construction of social infrastructure including railroads, communications etc. despite civil war and political instability. As a result, Chinese paleontology had experienced a rapid growth in both the number of paleontologists and paleontological research. Their research covered nearly all major animal phyla and plant divisions. Furthermore, their studies not only included taxonomic and biostratigraphic work, but also included discussions on the evolution of some specific taxonomic groups. Chinese paleontologists had also established a preliminary stratigraphic framework spanning from the Precambrian to the Quaternary. It is also notable that many of the first-generation paleontologists had some background in studying overseas or benefitted in some way from collaborating with some of the talented western paleontologists. However, despite the persistence of dedicated scientists, overall, the wars and social turmoil had slowed down the sometimes-fast development of Chinese paleontology.

## 4. From 1949 to 1977—The Expanding Stage of Chinese Paleontology

The founding of the People’s Republic of China in 1949 had two immediate impacts: first, the newfound social stability and the secondly, the cessation of academic exchange with western countries. Learning from and modeling after the former Soviet Union became a trend. China’s connection with countries other than the former Soviet Union was limited. For instance, Zhang Miman (Meemann Chang) spent a year visiting the Swedish Museum of Natural History in 1966 (Figure 3), but did not return there to receive her Ph.D. degree from Stockholm University until 1982.

To reboot its economy and social development following the civil war, the new Chinse government established the Chinese Academy of Sciences in 1950 which includes many different research institutes. Siguang Li was appointed the director of the Nanjing Institute of Paleontology in 1950, and the institute was officially founded in 1951, which also includes the Department of Vertebrate Paleontology in Beijing. The institute was based on staff merged from the paleontological group of the Institute of Geology, the Central Research Academy, the department of paleontology and the Cenozoic Research Laboratory (Beijing) of the National Geological Survey of China. In 1953, the Department of Vertebrate Paleontology, based in Beijing, was separated the from the Nanjing Institute of Geology and Paleontology and became affiliated directly with the Chinese Academy of Sciences. In 1957, the department was upgraded to an institute and assigned its current name, the Institute of Vertebrate Paleontology and Paleoanthropology (IVPP), in 1960 with Zhongjian Yang as its first director.

In 1959, the Nanjing Institute of Paleontology was renamed as the Nanjing Institute of Geology and Paleontology (NIGPAS) and has since focused on the study of invertebrate paleontology, paleobotany, and stratigraphy, in order to be distinct from the IVPP, but both are still affiliated with the Chinese Academy of Sciences (CAS) today. The two research institutes of the CAS have since become the major forces in paleontological study in China. It is also noteworthy that the two institutes were also able to accept graduate students and helped to educate many of the outstanding young paleontologists of the next generation.

In addition to the positions available at the two research institutes of the CAS (NIGPAS and IVPP), many other paleontologists were mainly employed at universities such as Peking University, Nanjing University, Northwest University, Beijing Institute of Geology (now the China University of Geoscience in Beijing and Wuhan), Beijing Institute of Mining and Technology (now the China University of Mining and Technology), Changchun Institute of Geology (now merged into Jilin University), and Chengdu Institute of Geology (now the Chengdu University of Technology) etc. They have educated many students in paleontology and stratigraphy, in addition to conducting extensive studies on invertebrates, plants and stratigraphy.

In addition, the Geological Academy of Sciences affiliated to the Ministry of Geology and other institutes affiliated to the later established Ministry of Oil have also recruited many staff members who are working on paleontology and stratigraphy, mainly to fulfil the tasks of geological surveying and exploring for geological resources.

In 1953, the journal *Acta Paleontologica Sinica* was established, with Zunyi Yang as chair of the editorial board. Minzhen Zhou, a 1950 Ph.D. from Lehigh University in the USA, helped create the journal *Vertebrata PalAsiatica* in 1956, with Zhongjian Yang as the chair of the editorial board. In 1957, Zunyi Yang and Yichun Hao, graduates from the National Southwestern Associated University, published the first textbook on paleontology for Chinese students. In the same year, Yichun Hao also published the first Chinese textbook on micropaleontology.

During this time, the extensive geological surveys conducted due to the nationwide need to domestically locate oil, coal, and other mineral resources stimulated a rapid growth in the number of paleontologists around the country. Micropaleontology was quickly developed, largely in response to the practical demand for experts in the field. While academic exchange between China and western countries had nearly stopped, a new generation of students were sent to the former Soviet Union to study paleontology. Among the graduates were Yichun Hao, Pinxian Wang, and Miman Zhang (Meemann Chang). Chinese paleontologists had been asked to edit books on categories of fossils from various regions of China and made progress on the study of fossils and stratigraphy of nearly all Phanerozoic ages of China. For instance, dinosaur excavations at Shandong, Sichuan, Inner Mongolia and Xingjiang have produced many previously unknown taxa from the Jurassic and Cretaceous. In addition, large scale expeditions were organized to investigate the geology of Tibet.

Despite the general practical need of the geological surveys to explore mineral resources, there was still some scientific consideration for the paleontological development in China thanks to the enduring visions of the first generation of Chinese paleontologists (Figure 4). For instance, Zhongjian Yang had outlined the focus and directions of the IVPP as “four origins and two deposits” in 1955. The four origins include the origins of vertebrates, tetrapods, and mammals, as well as humans, primates and cultures from the perspective of biology. The two deposits represent the fossil bearing red beds in southern China and the soil-like deposits in northern China from the perspective of geology. In 1958, Yang further proposed the goals of “three gaps to be filled” for the IVPP, i.e., the evolutionary gap, regional gap, and the stratigraphic gap. In the 1960s, the discovery of human fossils had been remarkable, including some new specimens of *Homo erectus*, and *Homo sapiens*.

International collaboration between China and the former Soviet Union is not only limited to the education of Chinese students by experts from the former Soviet Union. In 1959, in adherence to an agreement between the Academies of Sciences of the two countries, a joint expedition to Central Asia was formed (Figure 5), and was led on the Chinese side by Zhou Minzhen from the IVPP. Unfortunately, the joint expedition did not last long and was ended in 1960 due to the deterioration of the political relationship between the two countries.

During the Cultural Revolution (1966–1976), much like many other scientific disciplines, Chinese paleontology was seriously disrupted. In particular, the Chinese paleontological community was nearly completely isolated from the outside world. Despite the turmoil, field excavations continued to produce some exciting finds, and the empirical paleontological and stratigraphic work carried out during that time filled many gaps in our understanding of both the evolution of life and the stratigraphic record. In addition, micropaleontology witnessed a stage of rapid development due to its application in surveying and exploring for geological resources.

In summation, with the second generation of paleontologists entering the mainstage during the period from 1949 to 1977, the Chinese paleontological community was greatly expanded. On one hand, this was under the guidance of the first generation of paleontologists and their international vision. On the other hand, it was also impacted by the influence of the former Soviet Union and the growth of Chinese paleontology had been closely related to the national industrial requirement for extensive geological surveys and prospecting with a focus on the collection of fossils from across the country spanning from the Precambrian to the Quaternary. A preliminary biostratigraphic frame was established, and the study of the Chinese paleontologists covered nearly all the taxonomic categories of paleontology. Overall, Chinese paleontology had generally been geology-oriented, although the biological significance of some fossils had also been discussed to a degree.

## 5. From 1977 to 2020

In 1977, the National College Entrance Exam was reinstated after it had been abandoned for ten years and as a result many more talented students were able to study paleontology in college. A new generation of graduate from Chinese universities were able to be recruited into the NIGPAS, IVPP, and other research institutes as well universities, and they remain the backbone of Chinese paleontology today.

With the beginning of the open-door policy in 1978, the connection between China and western countries was able to be re-established. While some senior paleontologists were able to communicate with their international colleagues and had more and more visits from both sides, some Chinese students were able to pursue their degrees in western universities. Meanwhile, new ideas and the most recent developments including scientific methods in western countries such as cladistics and paleoecology began to be introduced into China [15,16].

Since the 1980s, Chinese paleontologists have been frequently invited to participate in the editing of the Treatise on Invertebrate Paleontology. In addition, Chinese paleontologists have published many books about their systematic work on various groups such as trilobites, brachiopods, graptolites, corals, bivalves, cephalopod, insects, conchostrans etc.

Starting in the late 1990s, Chinese paleontology began to gradually enter its “golden age”. First, some of the western educated students came back to China and were awarded enough funding for their research to establish their own research labs. Secondly, many domestically educated students graduated from colleges and joined research institutes such as the NIGPAS and IVPP. The third generation of paleontologists, or the “reform and open generation”, had the advantage of having a better grasp of the English language and were able to more easily learn new technique and methods. It is notable that some of the Chinese students chose to stay in western countries after earning their Ph.D. degrees continued their paleontological career, and they helped to foster collaborations between China and various western countries in order to both educate students and sponsor further collaborative projects. More international meetings were hosted in China and more Chinese paleontologists were able to participate in meetings outside China.

Starting in 2000, the scale of funding from the government for basic scientific research began to be increased. The National Natural Science Foundation, which was founded in 1986, has been growing steadily and has become the main source of support for Chinese paleontologists since the 1990s. For instance, over two dozen of the third and fourth generation of paleontologists have been supported by the Distinguished Young Scientist Fund, and paleontologists from the NIGPAS, IVPP, China University of Geosciences, and Northwest University have also been awarded the Innovation Research Group Fund to ensure that their research is being supported in the long term.

In addition to the funding from the NSFC, Chinese paleontologists had a chance to be awarded large grants from other government agencies to help organize a large group of paleontologists to work together on a major scientific endeavor. For instance, the Special Funds for Basic Research of China (“973” projects) by the Ministry of Sciences and Technology provided grants amounting to twenty million Chinese yuan for a five year’s project [17]. Since 2000, several paleontologists from the NIGPAS, IVPP, and the University of Geosciences (Wuhan, Beijing) have been awarded this grant. In addition, Chinese paleontologists have recently been able to secure other major grant, i.e., the Strategic Priority Research Program from the Chinese Academy of Sciences, to investigate the coevolution of life and paleoenvironment during major critical intervals of Earth’s history. Most recently, a 10-year multidiscipline project was awarded to a paleontologist by the NSFC. It is intended to study the Mesozoic terrestrial biota and its tectonic background from the perspective of Earth system science, and it has drawn participants from paleontology, geochemistry, geophysics, and sedimentology. Furthermore, even more paleontologists have been invited to participate in several other multidiscipline projects, e.g., the new Tibetan Plateau Expeditions.

The state key laboratory administrated by the Ministry of Sciences and Technology is another way to support basic scientific research in China. The State Key Laboratory of Paleobiology and Stratigraphy, based at the Nanjing Institute of Geology and paleontology, and the State Key Laboratory of Biogeology and Environmental Geology, based at the Chinese University of Geosciences were established in 2002 and 2012, respectively. These labs are annually fiscally supported not only in scientific research and collaboration, but also in the acquisition of new equipment.

Due to the increasing funding capacity available to the promising young generation of paleontologists, Chinese paleontology has been growing at an unprecedent rate, and their research scope and production have been greatly expanded, spanning from traditional paleontology and stratigraphy, to paleobiology and some newly developed fields. Since 2000, some students and postdocs from western countries started to join the Chinese institutions and universities, which have further fostered the international collaborations between China and its international community.

In traditional paleontology and stratigraphy, Chinese paleontologists have greatly extended their scope of excavations of fossils to nearly all areas of China, spanning from Precambrian to the Quaternary. As a result, the rate of discovery of new taxa in various biological groups has significantly increased. For instance, the rate of commonly accepted dinosaur species named from China now outnumbers that of any other country. Starting 2015, supported by the “Special Research Program of Basic Science and Technology of the Ministry of Science and Technology”, Chinese paleontologists launched an ambitious project to publish a complete series of *Palaeovertebrata Sinica*, which comprises three volumes and 23 fascicles on the taxonomy of all the vertebrate fossil species (nearly 10 thousand) in China. Currently, 13 fascicles have been published, and several more are nearly finished.

Among many of the discoveries, there are several world famous Lagerstätte are probably best known to the international paleontological community, i.e., the Neoprotozoic Wengan Biota (580 my) in Guizhou, Early Cambrian Chengjiang Biota (520 my) in Yunnan, the late Jurassic Yanliao Biota (160 my), and the Early Cretaceous Jehol Biota (125 my) in western Liaoning and neighboring areas, which have produced many world-known exceptionally preserved fossils bearing a lot evolutionary significant data on the early evolution of early life including the evolution of animals [18,19,20,21], origin of birds, and regarding the early evolution of mammals, birds, pterosaurs, and flowering plants etc. [22,23,24,25]. In addition, these discoveries have also drawn great attention from the media and public, hence increasing the profile of Chinese paleontology in popular scientific communities. In 2001, *Science* magazine published a special issue introducing the highlights of the discoveries and research done in Chinese paleontology.

Thanks to sufficient funding and the great efforts put into field investigations, many other Cambrian fauna have also been discovered, including the Neoprotozoic Lantian Biota (600 my) in Anhui [26] and the Miaohe Biota in Hubei, the terminal Ediacaran Shibantan Biota of Yangtze Gorges, South China [27], the Edicarian fauna in Guizhou [18,28], the Early Cambrian Burgess Shale-type fossil Lagerstätte the Qingjiang Biota [29], and the Middle Cambrian Burgess Shale-type fossil Lagerstätte the Lingyi Biota [30] (Figure 6), etc. Newly discovered Silurian fishes from Chongqing have produced many exceptionally preserved articulated jawed fished, among many other discoveries. Devonian plants from Yunnan and other regions produced unknown vascular plants [31]. The Permian Pompeii (about 300 my) preserved remarkable plants from coal in Wuhai, Inner Mongolia [32]. Many new species of Middle Triassic marine reptiles and fishes often represented by complete skeletons, including the earliest turtle, have also been discovered in Guizhou and Yunnan provinces [33]. The Miocene Hezheng Fauna has also produced abundant and diverse mammals and birds, sometimes with exceptional preservation of soft tissues and gut contents [34]. Lastly, the Miocene Zhangpu Biota from Fujian represents another amber treasure trove, recording a diverse fauna and flora of the tropical forest ecosystem [35].

Chinese paleontologists have also made significant contributions to several of the major evolutionary issues, such as the Cambrian explosive radiation and its background [36], the origin of animals [37], the study on the Great Ordovician Radiation and the end-Ordovician mass extinction [38,39], the study of early vertebrates [40], the P-T extinctions [41], the biodiversity changes based on big data [42], the origin of birds and their flight [23,43], the early evolution of Mesozoic mammals including middle ears [22,44], the Mesozoic insect–plant coevolution [45], sexual selection [46,47], the interaction of the Cenozoic biota and flora with the uprising of the Tibet Plateau [48,49], the evolution and dispersal of modern and archaic humans in China, etc. [50,51,52].

In the study of stratigraphy, Chinese paleontologists have now witnessed the ratification of 11 global boundary stratotype sections and points (GSSP, or the so-called golden spikes) in China, including the GSSP of the boundary between the Paleozoic and Mesozoic in Zhejiang Province [53,54]. From the study of terrestrial strata, the Paleogene and Neogene biochronologic frameworks in China were preliminarily established [55]. New integrative stratigraphy and timescales for 13 geological periods (Ediacaran–Quaternary) in China were published in the special issue of SCIENCE CHINA Earth Sciences co-edited by Shen and Rong [56]. This research summarized the latest advances in stratigraphy and timescale and discussed the correlation among different blocks in China with international timescales.

High-resolution stratigraphic studies have also been helpful in oil and gas explorations as well as marine geology. Chinese paleontologists, particularly invertebrate paleontologists and micropaleontologists, continue to have work in close collaboration with oil companies for the prospecting of oil and gas resources in which the Silurian division by graptolites has played a key role [57].

Paleogeographic, paleoecological, and paleoenvironmental research has also been active. For instance, paleogeographic and paleoecological reconstruction has been useful in studying the controversial rising history of the Tibetan Plateau [49,58], and the paleobiogeographic and paleogeographic evolution of blocks in the Qinghai–Tibet Plateau [59].

It is notable that the geochronological progress in China has made a great contribution to the establishment of stratigraphic frame, which is critical for the discussion of various geological and evolutionary questions. In particular, the dating of the volcanic ashes interbedded in the terrestrial sediments provided a rare chance for the precise correlation of deposits as well as the fossils in them [60,61,62,63]. Furthermore, the precise dating of the fossil-bearing deposits enabled us to better relate the evolution of the Jehol Biota to the major tectonic background [64,65]. Exciting dating results have been obtained from deposits ranging from the Edicaran to the Quaternary, thus providing a solid ground for discussing the interactions between the evolution of different forms of life and their geological and paleoenvironmental background.

Geobiology, derived from the study of geomicrobiology, has also been developing quickly in China [66], and shows a promising future as it can better bridge the geological and biological processes in Earth’s history by taking advantage of the latest advances in the study of geochemistry and molecular biology. It is also noteworthy that an institute called the Institute of Geo-Biology was founded in Beijing in 1940, with P. Teilhard de Chardin as the honorary president and zoologist P. Leroy as the director. It was based on the collections and laboratories of the Huangho-Paiho Museum (now the Tianjin Natural History Museum) founded by F. Licent in 1915.

The application of the modern genomic technique to the study of fossils has quickened the development of the discipline of molecular paleobiology, particularly regarding the study of fossil DNA. In the past decade, Chinese paleontologists have succeeded in extracting the DNA sequences from bones of various fossils of *Homo sapiens*, as well that of Denisovan from sediments [67,68]. The rapid progress in this area has also been well connected with the study of archaeology and will certainly make a big impact on the study of the early history of modern humans in China as well as the cultural exchanges between various human populations in the past 100 ka [69,70]. In addition, the extraction of fossil DNA has been increasingly often used in the study of Quaternary fossil mammals such as the Giant Panda.

The study of molecular paleobiology is not limited to the study of fossil DNA. In fact, the study of fossil proteins in deep time has also shown a promising future. A recently published work on the paleoproteomics of the mysterious primate *Gigantopithecus*, dating back 1.9 million years, provided some interesting phylogenetic information [71]. It is also notable that Chinese paleontologists have also been working hard on the identification and detailed analysis of keratin from fossil feathers of dinosaurs and birds from the Jurassic and Cretaceous [72].

The rich paleontological and stratigraphic resources that remain to be further investigated and the unique geological history of the Chinese continent provide numerous chances for the future development of Chinese paleontology, particularly when paleontology is regarded as an integrated part of the Earth System. For instance, the collision between the Indian plate and Eurasian plate and the subsequent rise of the Tibetan Plateau in the Cenozoic has been one of the major tectonic events that has made an important impact on the geography, climate, and history of China. The study of the evolution of life, including human evolution in the Tibetan Plateau against its unique geological background, is expected to produce more exciting multidiscipline results. The subduction of the paleo-Pacific plate towards the Eastern Asia during the late Mesozoic in East Asia has been regarded as related to the evolution of the Yanliao and Jehol Biota, and the study of the control of deep tectonic activities on the surface geological processes and its impact on the evolution of terrestrial life has now drawn attention from paleontologists, geologists, and geochemists.

Technical applications, such as synchrotron, have also played a key role in the study of paleontology in China [73,74,75]. Several Chinese institutes and paleontological labs in universities have now been equipped with high resolution CT in addition to SEM, TEM, etc. The Synchrotron facilities in Shanghai, Beijing, and Hefei as well as those in Taiwan have also been used by Chinese paleontologists, bringing forward more opportunities and research directions for the younger generation of paleontologists [76,77,78].

The NIGPA has constructed an impressive Geobiodiversity Database (GBDB) thanks to the longstanding support of the State Key Laboratory of Paleobiology and Stratigraphy and other major projects. Recently, a high-resolution summary of Cambrian to Early-Triassic marine invertebrate biodiversity curves with an imputed temporal resolution of 26 ± 14.9 thousand years was published based on the Geobiodiversity Database (GBDB), which used quantitative data from 11,000 marine fossil species collected from more than 3000 stratigraphic sections in China [42].

Benefitted by the rapid economic growth and the need for popular science education, many new museums of natural history or geology have been constructed in China, which have created more chances for the employment of graduates of paleontology. While the NIGPAS and IVPP remain the two biggest paleontological centers in China, a few universities, e.g., the Chinese University of Geosciences, Peking University, Nanjing University, Northwest University, Lanzhou University, and the Geological Academy of Sciences remain to have a strong paleontological program. New paleontological programs have been established and are growing in several other universities, such as the Yunnan University, Sun Yat-Sen University, Capital Normal University in Beijing, Shenyang Normal University in Liaoning, Hefei University of Technology in Anhui, and Lingyi University in Shandong, etc. In addition, paleontologists are also active in some of the biological institutes, e.g., the Xishuangbanna Tropical Botanical Garden, Institute of Botany of the CAS, and many provincial museums.

Despite the remarkable progress made during the past three decades, the advancement of Chinese paleontology is also held back by several challenges. For instance, the illegal collecting and marketing of vertebrate fossils remain unsolved issues, while scientific collecting often meets great difficulty mainly due to the immature administrative management of fossil resources and lack of local interest.

In sum, due to over 40 years of reform and the open-door policy of China in addition to the rapid growth of the nation’s economy, paleontology in China has largely merged into the global paleontological community and became a major force that constantly produces exciting new discoveries of fossils that arouse public interest. Furthermore, it has contributed important evidence to ongoing evolutionary research, which has added to our understanding of the tree of life in deep time. The integration of paleontology with biological and geological sciences has indisputably proved its importance.

## 6. Future Directions and Challenges

The new generations of paleontologists in China, who have benefited from the success of their predecessors against the background of digital age and several decades of the fast growing of Chinese economy, seem to be more optimistic and confident in using new technology and methods in paleontological studies. In addition, they are more willing and prepared to participate in multidisciplinary research that often involves a background in geochemistry as well as in molecular and developmental biology. More urgent global changes also encourage them to pay more attention to the interactions between life evolution and paleoenvironment during deep time.

During the digital age, more attention has been paid to the construction of Big Data and the application of AI technology to paleontological study. Paleontological data will probably be better integrated with geochronological and geochemical data, which may help produce more interesting results regarding paleobiodiversity and paleogeography, as well as the impact of paleoenviroment, on the evolution of life on Earth.

Considering the population of China, in the future, there will be even more museums or universities with paleontology programs to increase the employment chances for paleontology students and arouse more public interest in this discipline. Young paleontologists can find their positions either in the department of Earth sciences or biology. Their expertise in the history of the Earth and the evolution of life on Earth will be a necessity not only for students of Earth sciences and biological majors, but also for any students who may be interested in the history of the Earth and life, or the theory of evolution (Figure 7).

With the growing need for science communication in order the increase the scientific literacy of the public, Chinese paleontologists should be optimistic and enthusiastic about the great potential of paleontology in attracting both kids and adults. It is important to note that when natural sciences are further integrated with liberal sciences, the knowledge gained in paleontology and evolutionary biology will be beneficial to a wide range of readers and future scientists.

However, challenges remain in Chinese paleontology. The practical philosophy of Chinese culture has hindered the development of basic scientific research. Some of the students chose paleontology because they saw it as an advantageous career rather than out of their personal interest. The NIGPAS and IVPP, as CAS’s two institutes, seem to be under some pressure to focus on institutional organized projects, rather than small individual projects born out of pure curiosity.

Due to historical and cultural reasons, it seems that Chinese paleontologists are shy to propose new hypotheses, attempt contribute more to evolutionary theory from paleontological evidence, or to try to integrate paleontological data with biological data. Cross-discipline interaction is never easy, and it only happens when the scientific culture is suitable for scientists to focus on purely scientific pursuits. Yu recently provided an interesting account of the social, cultural, and disciplinary factors that influenced the reception and appropriation of Darwinism by China’s first-generation paleontologists [79]. With the development of young generations of paleontologists in the new century, it is hoped that Chinese paleontology will continue to make more contributions to world’s paleontological community. This is accomplished not only by making more exciting discoveries, but also by engaging in new studies that enrich our understanding of the evolutionary mechanisms of life on Earth and provide more clues to our understanding of the impact of global changes in biodiversity on human evolution.

## Figures and Tables

**Figure 1 biology-11-01104-f001:**
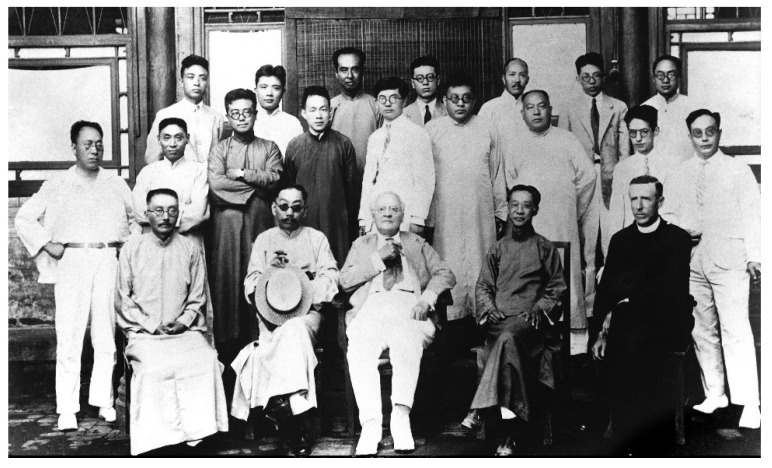
Founders of Chinese geology and paleontology gathering at Amadeus W. Grabau’s home in Beijing in 1935. Front row, from the left: Hongzhao Zhang, Wenjiang Ding, Amadeus W. Grabau, Wenhao Weng, and Pierre Teilhard de Chardin. Middle row, from the left: Zhongjian Yang, Zanheng Zhou, Jiarong Xie, Guangxi Xu, Yunzhu Sun, Xiechou Tan, Shaowen Wang, Zanxun Yin, and Fuli Yuan. Back row, from the left: Zuolin He, Hengshen Wang, Zhuquan Wang, Yuelun Wang, Huanwen Zhu, Rongseng Ji, and Jianchu Sun (credit to IVPP).

**Figure 2 biology-11-01104-f002:**
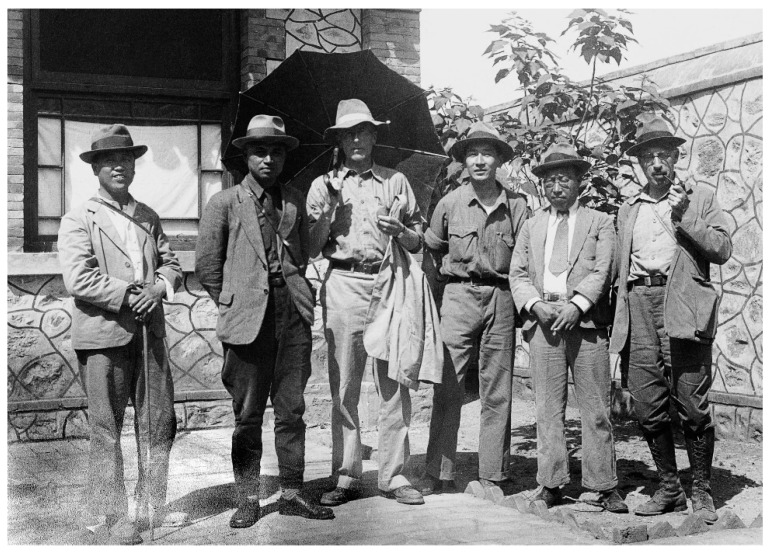
Paleontologists in Choukoudian, Beijing in 1934. From the left: Wenzhong Pei, Shiguang Li, Pierre Teilhard de Chardin, Meinian Bian, Zhongjian Yang, and George B. Barbour (credit to IVPP).

**Figure 3 biology-11-01104-f003:**
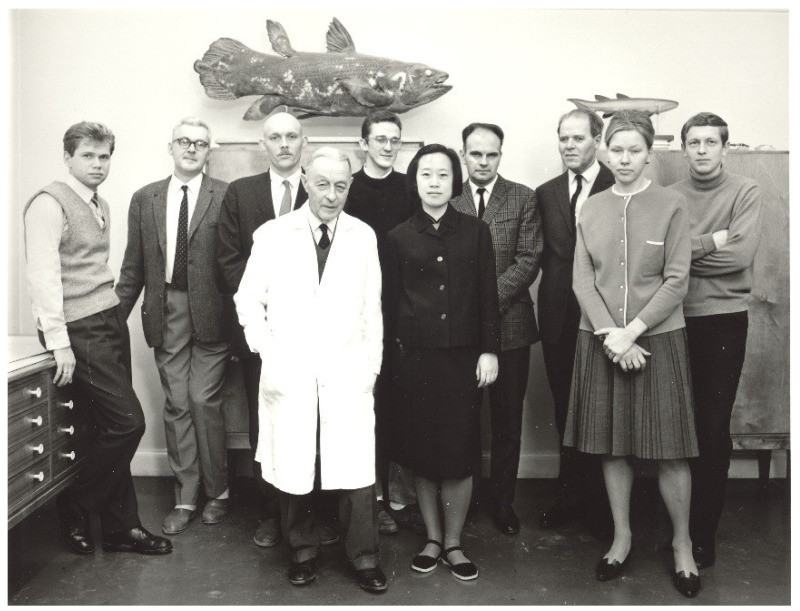
Miman Zhang and her colleagues in Stockholm, Sweden in 1966. Front row, front the left: Erik Stensiö, Miman Zhang, E. Mark-Kurik. Back row, from the left: Hans-Peter Schultze, Tor Örvig, Hans Bjerring, Gareth Nelson, Ray Thorsteinsson, Erik Jarvik, and Hans Jessen (credit to IVPP).

**Figure 4 biology-11-01104-f004:**
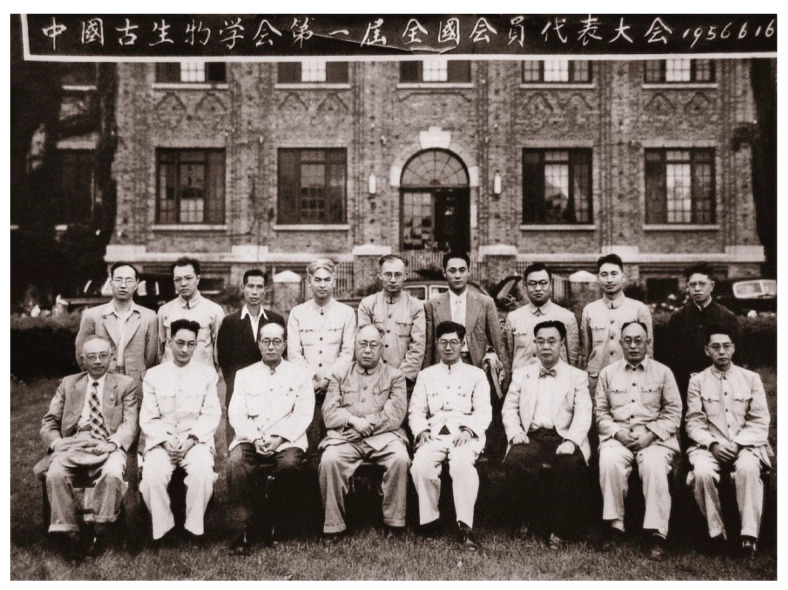
Group photo of officials elected at the first National Congress of the Chinese Paleontological Society in 1956. Front row, front the left: Senxun Yue, Zanxun Yin, Xiaohe Zhou, Zhongjian Yang, Yunzhu Sun, Xingjian Si, Jinke Zhao, and Zunyi Yang. Back row, from the left: Hongzhen Wang, Zhiwei Gu, Yuanren Qu, Shicheng Huo, Ren Xu, Minzhen Zhou, Yu Wang, Enzhi Mu, and Longqing Chang (credit to NIGPAS). The Chinese characters means the photo was taken at the first National Congress of the Chinese Paleontological Society on June 16^th^, 1956.

**Figure 5 biology-11-01104-f005:**
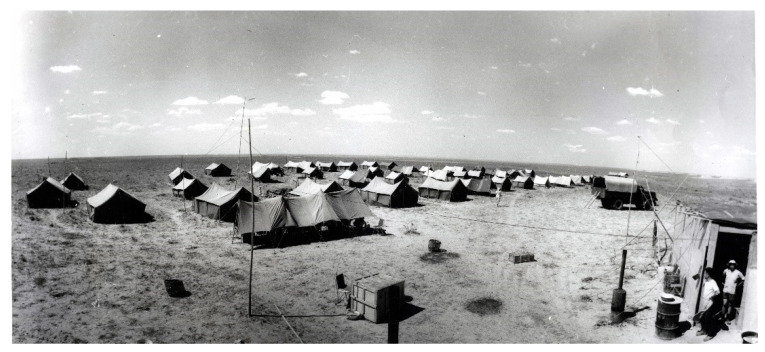
Field camp of the Sino–Soviet Union joint expedition in Inner Mongolia in 1959 (credit to IVPP).

**Figure 6 biology-11-01104-f006:**
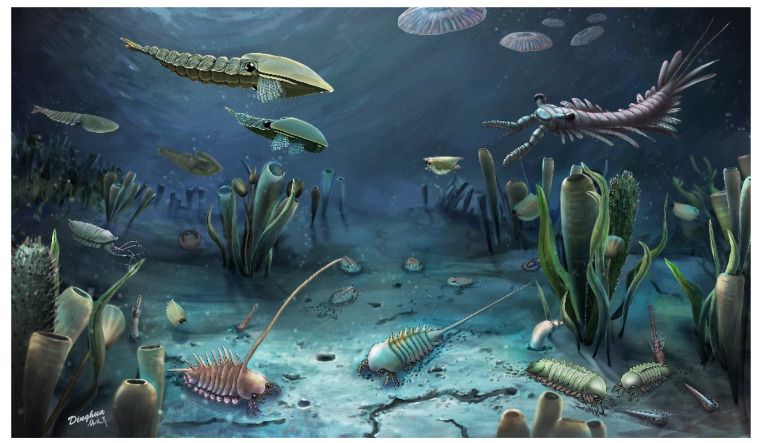
Reconstruction of the latest discovered Middle Cambrian Lagerstätte Lingyi Biota in Shandong (credit to Fangcheng Zhao).

**Figure 7 biology-11-01104-f007:**
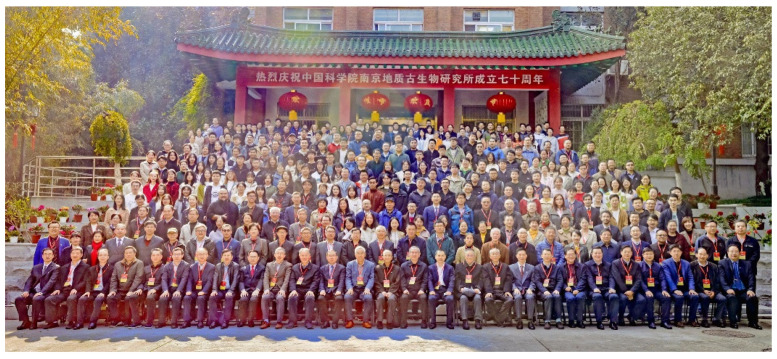
Group photo of participants of the conference celebrating the 70th anniversary of the Nanjing Institute of Geology and Paleontology in Nanjing in 2021 (credit to NIGPAS). The Chinese characters means a warm congratulation to the 70th anniversary of the Nanjing Institute of Geology and Paleontology of the Chinese Academy of Sciences.

## Data Availability

Not applicable.

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
