# Peer review of "The Rising of Paleontology in China: A Century-Long Road"

_biology, 2022, doi:10.3390/biology11081104_

Round 1
Reviewer 1 Report
Paleontology in China: a century’ road from 1920 to 2020
1. Page 1, Introduction section, line 4 from above: Please replace “Intervals” with “periods”.
2. Page 1, Section “2. Before 1920—Collecting …”, line 1 from above: “animals and plants” –perhaps and fungi …
3. Page 1, Section “2. Before 1920—Collecting …”, lines 7-11 from above: Some of this first paleontological books could be cited.
4. Page 1, Section “2. Before 1920—Collecting …”, line 3 from below: China is enormous! Please, give at least the name of province of this first explorations.
5. Page 2,, line 2 from below: Please, explain the abbreviation “GSSPs”.
6. Page 3, line 7 from above: “degrees from France” – Please, give names of cities, universities, as you have given for British and American ones.
7. Page 3, line 10 from below: “Western organizations”, or “organizations of Western countries”.
8. Page 3, line 6 from below: R. C. Andrews’ paleontological expeditions have discovered some of the earliest placental mammals.
9. Page 4, line 9 from below: “Austria” – “Austrian”.
10. Page 5, line 9 from above: A suggestion: It could be included a copy of the cover of the 1st issue of the emblematic journal “Palaeontologia Sinica”, as the article deals with history of Chinese paleontology.
11. Page 5, line 2 from below: “a lot of fishes, …” - “a lot of fossils of fishes, …”
12. Page 6, line 2 from above: to delete dot in “(Chow 1923). representing”.
13. Page 6, line 16 from below: Change “Lunfeng dinosaur fauna” to “Lufeng dinosaur fauna”.
14. Page 6, line 3 from below: “all major animal phyla and plants.” - “all major animal phyla and plant divisions.”
15. Page 9, line 3 from above: Please replace “between the Academy of Sciences of the two countries” with “between the academies of sciences of the two countries”.
16. Page 9, line 9 from below: “time 1949-1977” – “period 1949-1977”.
17. Page 9, line 6 from below: “to learn paleontology” - “to study paleontology”.
18. Page 11, line 12 from below: “etc” - “etc.”.
19. Page 12, line 17 from below: “modern and archaic human” - “modern and archaic humans”.
20. Page 13, line 16 from below: “Chinese continents”.
21. Page 15, line 1 from below: “earth” – “Earth”.
Author Response
Reply to reviewer 1
- Page 1, Introduction section, line 4 from above: Please replace “Intervals” with “periods”.
Changed.
- Page 1, Section “2. Before 1920—Collecting …”, line 1 from above: “animals and plants” –perhaps and fungi …
Changed.
- Page 1, Section “2. Before 1920—Collecting …”, lines 7-11 from above: Some of this first paleontological books could be cited.
Several new references have been added.
- Page 1, Section “2. Before 1920—Collecting …”, line 3 from below: China is enormous! Please, give at least the name of province of this first explorations.
Changed and see more introduction in other parts of the text.
- Page 2,, line 2 from below: Please, explain the abbreviation “GSSPs”.
Added.
- Page 3, line 7 from above: “degrees from France” – Please, give names of cities, universities, as you have given for British and American ones.
To be consistent, I have decided not to mention names of cities, universities.
- Page 3, line 10 from below: “Western organizations”, or “organizations of Western countries”.
Changed.
- Page 3, line 6 from below: R. C. Andrews’ paleontological expeditions have discovered some of the earliest placental mammals.
Added.
- Page 4, line 9 from below: “Austria” – “Austrian”.
Corrected.
- Page 5, line 9 from above: A suggestion: It could be included a copy of the cover of the 1stissue of the emblematic journal “Palaeontologia Sinica”, as the article deals with history of Chinese paleontology.
Unfortunately, I cannot find a copy of the cover of the 1st issue of the emblematic journal “Palaeontologia Sinica”.
- Page 5, line 2 from below: “a lot of fishes, …” - “a lot of fossils of fishes, …”
Changed.
- Page 6, line 2 from above: to delete dot in “(Chow 1923). representing”.
Changed.
- Page 6, line 16 from below: Change “Lunfeng dinosaur fauna” to “Lufeng dinosaur fauna”.
Changed.
- Page 6, line 3 from below: “all major animal phyla and plants.” - “all major animal phyla and plant divisions.”
Changed.
- Page 9, line 3 from above: Please replace “between the Academy of Sciences of the two countries” with “between the academies of sciences of the two countries”.
Changed.
- Page 9, line 9 from below: “time 1949-1977” – “period 1949-1977”.
Changed.
- Page 9, line 6 from below: “to learn paleontology” - “to study paleontology”.
Changed.
- Page 11, line 12 from below: “etc” - “etc.”.
Changed.
- Page 12, line 17 from below: “modern and archaic human” - “modern and archaic humans”.
Changed.
- Page 13, line 16 from below: “Chinese continents”.
Changed.
- Page 15, line 1 from below: “earth” – “Earth”.
Changed.
Reviewer 2 Report
This is a welcome contribution to the history of paleontological studies in China. It moves from an 'institutional' viewpoint, i.e., one that focus on persons, careers and academic institutions rather than on more strictly scientific contents. In my opinion, this special focus of the paper may be better clarified starting from the article's title.
The English text is always intelligible, though in need of a check by some native English speaker. Typos and grammar oversights are indeed widespread (though, as said, they do not compromise the understandability of the manuscript).
The first part of this review manuscript suffers from a clear lack of referencing to primary literature - it seems to me that a single work is cited in the first five pages!
The last part of the paper, i.e. that focusing on the 1970s onward, may somewhere seem a bit celebrative of the government's support of (and achievements in) palaeontological research.
Other minor comments:
-Page 2, there is a weird crasis between Teilhard de Chardin and Grabau (to be fixed). Moreover, the name of Pierre Teilhard de Chardin is abbreviated in various different ways all along the main text, and that is not very helpful.
- P. 6, "China's War of Resistance against Japanese Aggression", why not "Second Sino-Japanese War"? That's how that war is mostly known outside China and Japan...
- P. 6, Zhongjian Yang or Young? Please, try and use homogeneous romanizations.
P. 12, the last sentence is incomplete.
P. 13, the author mentions geobiology. I thinkt it would be important to notice that an Institute of Geobiology was founded in Peking in the early 1940s, having P. Teilhard de Chardin as honorary president and P. Leroy as director. If memory don't fail me, the same Institute published the first journal devoted to the field of Geobiology.
Author Response
Reply to reviewer 2
This is a welcome contribution to the history of paleontological studies in China. It moves from an 'institutional' viewpoint, i.e., one that focus on persons, careers and academic institutions rather than on more strictly scientific contents. In my opinion, this special focus of the paper may be better clarified starting from the article's title.
Thanks, the title has been changed.
The English text is always intelligible, though in need of a check by some native English speaker. Typos and grammar oversights are indeed widespread (though, as said, they do not compromise the understandability of the manuscript).
The manuscript has now been read by a native English speaker and extensive edited.
The first part of this review manuscript suffers from a clear lack of referencing to primary literature - it seems to me that a single work is cited in the first five pages!
Several new references have been added.
The last part of the paper, i.e. that focusing on the 1970s onward, may somewhere seem a bit celebrative of the government's support of (and achievements in) palaeontological research.
It has been a bit tuned down.
Other minor comments:
-Page 2, there is a weird crasis between Teilhard de Chardin and Grabau (to be fixed). Moreover, the name of Pierre Teilhard de Chardin is abbreviated in various different ways all along the main text, and that is not very helpful.
Fixed.
- P. 6, "China's War of Resistance against Japanese Aggression", why not "Second Sino-Japanese War"? That's how that war is mostly known outside China and Japan...
Changed.
- P. 6, Zhongjian Yang or Young? Please, try and use homogeneous romanizations.
Checked.
- 12, the last sentence is incomplete.
Fixed.
- 13, the author mentions geobiology. I thinkt it would be important to notice that an Institute of Geobiology was founded in Peking in the early 1940s, having P. Teilhard de Chardin as honorary president and P. Leroy as director. If memory don't fail me, the same Institute published the first journal devoted to the field of Geobiology.
Thanks, added.
Reviewer 3 Report
I beleive it is an importnat and very interesting contribution to the history of paleontology, which may be published in its present form.
Author Response
Thanks